# Primary Cutaneous Multifocal Indolent CD8+ T-Cell Lymphoma: A Novel Primary Cutaneous CD8+ T-Cell Lymphoma

**DOI:** 10.3390/biomedicines11020634

**Published:** 2023-02-20

**Authors:** Tina Petrogiannis-Haliotis, Kevin Pehr, David Roberge, Ryan N. Rys, Yury Monczak, Gizelle Popradi, Lissa Ajjamada, Naciba Benlimame, Christiane Querfeld, Nathalie Johnson, Hans Knecht

**Affiliations:** 1Division of Pathology, Jewish General Hospital, McGill University, Montréal, QC H3T 1E2, Canada; 2Division of Hematology, Department of Medicine, Jewish General Hospital, McGill University, Montréal, QC H3T 1E2, Canada; 3Division of Dermatology, Department of Medicine, Jewish General Hospital, McGill University, Montréal, QC H3T 1E2, Canada; 4Département de Radiologie, Radio-Oncologie et Médicine Nucléaire, Centre Hospitalier de l’Université de Montréal (CHUM), Montréal, QC H2W 1T8, Canada; 5Lady Davis Institute for Medical Research, Jewish General Hospital, McGill University, Montréal, QC H3T 1E2, Canada; 6Division of Hematology, Department of Medicine, Royal Victoria Hospital, McGill University, Montréal, QC H4A 3J1, Canada; 7Division of Dermatology, City of Hope, Duarte, CA 91010, USA

**Keywords:** primary cutaneous T-lymphoma, multifocal, indolent, CD8+, flow cytometry, cell sorting, immunohistochemistry, molecular fingerprinting, NGS

## Abstract

We report the case of a patient who was referred to our institution with a diagnosis of CD4+ small/medium-sized pleomorphic lymphoma. At the time, the patient showed a plethora of lesions mainly localizing to the legs; thus, we undertook studies to investigate the lineage and immunophenotype of the neoplastic clone. Immunohistochemistry (IHC) showed marked CD4 and CD8 positivity. Flow cytometry (FCM) showed two distinct T-cell populations, CD4+ and CD8+ (+/− PD1), with no CD4/CD8 co-expression and no loss of panT-cell markers in either T-cell subset. FCM, accompanied by cell-sorting (CS), permitted the physical separation of four populations, as follows: CD4+/PD1−, CD4+/PD1+, CD8+/PD1− and CD8+/PD1+. TCR gene rearrangement studies on each of the four populations (by next generation sequencing, NGS) showed that the neoplastic population was of T-cytotoxic cell lineage. IHC showed the CD8+ population to be TIA-1+, but perforin- and granzyme-negative. Moreover, histiocytic markers did not render the peculiar staining pattern, which is characteristic of acral CD8+ T-cell lymphoma (PCACD8). Compared to the entities described in the 2018 update of the WHO-EORTC classification for primary cutaneous lymphomas, we found that the indolent lymphoma described herein differed from all of them. We submit that this case represents a hitherto-undescribed type of CTCL.

## 1. Introduction

Primary cutaneous lymphomas (CL) constitute the second largest group of extranodal lymphomas. The majority of CLs are of T-cell lineage (CTCL) [1,2,3]. Of these, most represent neoplastic proliferations of T-helper memory cells, with a CD3+ CD4+ CD45RO+ immunophenotype. CTCLs of T-cytotoxic cell lineage are uncommon. They comprise subcutaneous panniculitis-like T-cell lymphoma, extranodal NK/T-cell lymphoma and nasal type and rare CTCL subtypes, which include primary cutaneous gamma-delta T-cell lymphoma and CD8+ aggressive epidermotropic CD8+ cytotoxic T-cell lymphoma. Such lymphomas may show aggressive clinical behavior, with systemic involvement. Similarly to other rare CTCL subtypes, primary cutaneous CD4+ small/medium T-cell lymphoproliferative disorder (CSMP-TCL/LPD) presents with a solitary lesion on the head, neck or upper trunk, represents a neoplastic proliferation of lymphocytes that are of alpha-beta T-cell lineage, have a CD3+ CD4+ CD8- CD279/PD-1+ (negative for cytotoxic proteins) immunophenotype and show indolent behavior.

The family of CD8+ CTCL was recently further enriched by the recognition of a distinct type of indolent CD8+ lymphoma, called primary cutaneous acral CD8+ TCL (CD8+ ACTL). CD8+ ACTL presents with solitary or, rarely, bilateral single nodular lesions at acral sites, namely the ears, face and feet [4], and this was confirmed and extended in a multicenter study [5]. These CD8+ neoplastic CD3+ lymphocytes are of alpha-beta T-cell lineage, are CD4- and express a nonactivated cytotoxic phenotype which is positive for TIA-1, but negative for other cytotoxic cell markers. By CD68 immunostaining, we found that they show a characteristic perinuclear dot-like staining pattern that is different from that observed in normal histiocytes [6].

As novel, including targeted, therapies emerge, it becomes increasingly important to accurately subclassify these rare T-cell lymphomas, including on the genetic level, as their morphological and immunophenotypic features often overlap [7].

Herein, we present a case of a hitherto-undescribed form of CD8+ CTCL that is characterized by multifocality and indolent clinical behavior. Our case bears some similarity to several previously described types of CTCL, particularly CD8+ ACTL. However, the clinical presentation and histological features vary significantly from the latter and other types of CTCL.

## 2. Materials and Methods

Clinical photographs of the patient’s cutaneous lesions (Figure 1) ware taken by a medical grade camera.

Immunohistochemical (IHC) studies: All IHC studies were performed on formalin-fixed paraffin-embedded (FFPE) tissue using commercially available antibodies, as summarized in Appendix A.

Fluorescence-activated cell sorting (FACS): FACS was carried out on a single cell suspension of mononuclear cells pooled from multiple lesions (histologically represented in Figure 2 and Figure 3). Briefly, cells were stained for CD45, CD3, CD19, CD4, CD8 and CD279/PD-1 (BD Biosciences). Gating was performed using FACSDiva Software (version 8.0.2, see Figure 4 for gating strategy). Sorting was carried out using a FACSAria III flow cytometer (BD Biosciences) with a 70 µm setup. Four populations were isolated based on T-cell subset and PD-1 expression. They are designated as follows: CD4+/PD1-, CD4+/PD1+, CD8+/PD1- and CD8+/PD1+ (Table 1 and Figure 4).

T-cell receptor (TCR) clonality studies consisted of an analysis to detect the presence of a predominant clonal TCR gene fragment, indicating the presence of a clonal population of T-lymphocytes. Sequencing of the T-cell receptor gamma (TRG) and beta (TRB) genes using next-generation sequencing (NGS) technology was based on the BIOMED-2 protocol (InVivoScribe LymphoTrack Assays, reagents and analysis software) on the Illumina MiSeq platform. The assay involved amplicon-based DNA library preparation followed by bi-directional NGS sequencing of all amplifiable TRG and TRB V-J rearrangements, using multiplex primers targeting conserved sequences in variable (V) regions, in combination with the joining (J) region of both TRG and TRB genes. The TRG and TRB sequences were analyzed using the InVivoScribe LymphoTrack IG MiSeq [8,9,10].

## 3. Results

### 3.1. Clinical History

A 44-year-old Caucasian female was referred to our cutaneous lymphoma clinic in February 2017, with a diagnosis of primary cutaneous CD4+ small/medium-sized pleomorphic T-cell lymphoma (CSMP-TCL). At the time, she had a 2-year history of an increasing number of red-purple papules and small nodules, 0.25–2.0 cm in size, initially only on her lower extremities, although specifically not on her feet. They were neither painful, nor tender, nor pruritic. Individual lesions would initially grow quickly over a few months, but rarely change in size thereafter. When we met her, she had more than 200 such lesions. She was otherwise in good health, on no medications, displayed no systemic symptoms and had no palpable lymph nodes or hepatosplenomegaly. Peripheral blood was negative for clonal T-cell receptor (TCR) gene rearrangement.

We noticed that this clinical picture was not in keeping with the referring diagnosis of CSMP-TCL, which commonly presents with only one lesion, usually on the head. Moreover, immunohistochemistry (IHC) displayed marked CD4+ and CD8+ staining, inconsistent with the described cases of CSMP-TCL/LPD and making it impossible to identify the neoplastic population. Gene rearrangement studies were consistently positive for clonal TCR gene rearrangement, including TRG and TRB.

Even though the diagnosis was not definitive, we felt the need to move forward with treatment. We administered focal, superficial single-dose (7 Gy) palliative radiation treatment to several physically troublesome lesions, with surprisingly disappointing results. We then attempted isotretinoin, up to 80 mg per day, without any benefit (bexarotene has limited availability in Canada, but our group has had excellent results using other systemic retinoids, including isotretinoin). Following that, we attempted photochemotherapy (psoralen + ultraviolet A, also known as PUVA) for 12 weeks, then MTX at 35 mg weekly, but neither was effective. Interferon (IFN) alpha-2b was started at a modest dose of 2.5 million units (subcutaneously, three times/week), but stopped within 3 weeks because of myalgias, cephalgia and fatigue. Concomitantly, mechlorethamine 0.01% ointment was started on the left lower extremity only (at patient’s request, with only one site initially treated to test tolerability), which was stopped after a few weeks because of its irritant nature. The patient declined to retry at a lower concentration.

Biopsies were repeated 3 times during the next 18 months, all during pauses in treatment of at least 4 weeks and all showing similar histologic results.

A CT of the chest–abdomen–pelvis, and later a PET-CT, were both negative for systemic involvement. In particular, the PET-CT showed no increased FDG uptake in the skin, including known disease sites. By 2021, the number of lesions had increased by approximately 20%, including, at this point, some on the ear, but still predominantly on the legs (Figure 1). Fewer than 10% of the older lesions had increased in size, and only a few of those significantly. The clinical history, biopsies and treatment regimens are summarized in Table 1.

**Figure 1 biomedicines-11-00634-f001:**
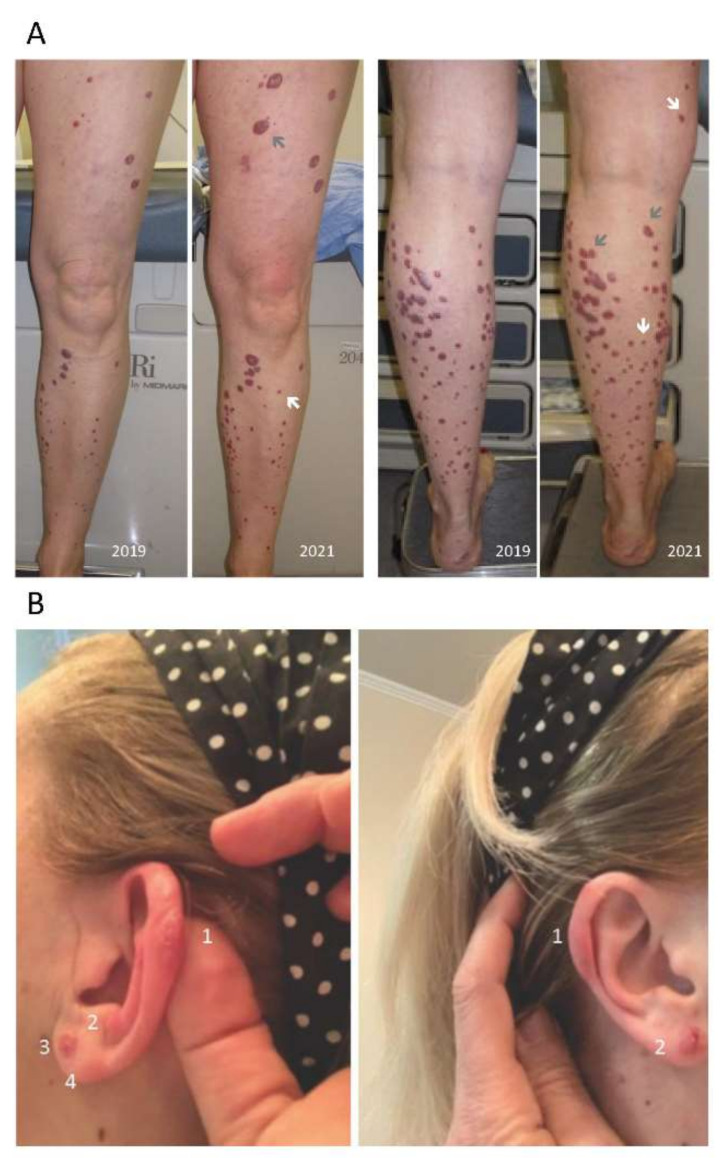
Clinical features. (**A**) Upper panels show the patient’s legs, in frontal and posterior views, in 2019 and 2021. One subungual lesion occurred (not shown). The leg lesions showed progression over 2 years, with some new lesions (white arrows) and some growing lesions (grey arrows). (**B**) Lower panels show the patient’s ears in 2021. Multiple lesions occurred on both ears (left, 4 lesions; right, 2 lesions, clearly distinguishable).

**Figure 2 biomedicines-11-00634-f002:**
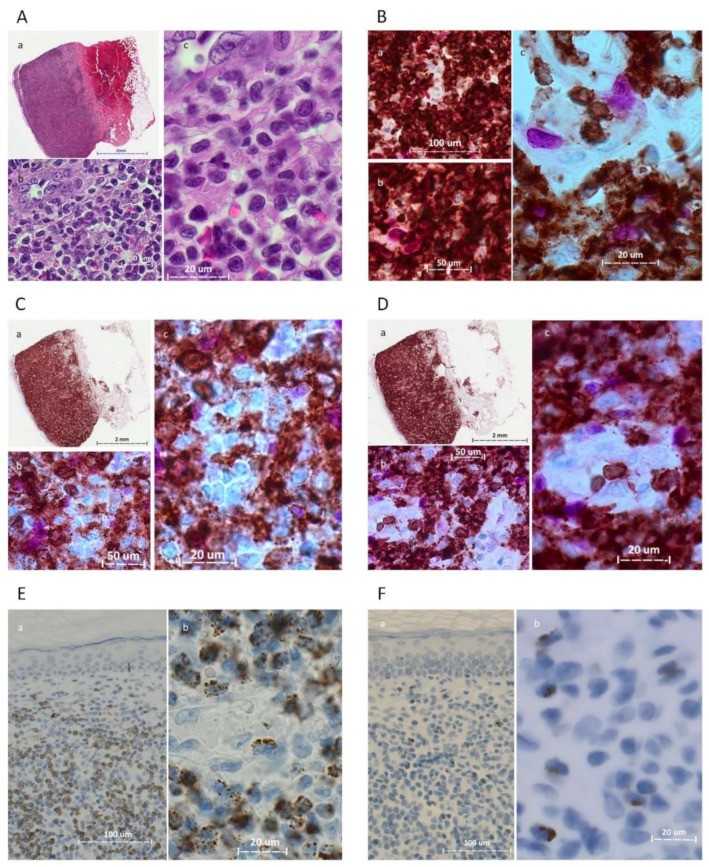
Photomicrographs of the 2021 skin biopsy. (**A**) Hematoxylin and eosin (H&E), Panel a, low magnification (1×, Aperio system); panels b and c, 40× and 63× oil (ZEISS Axioscope.5). (**B**) CD3 (brown)/Ki-67 (magenta) dual stain; panels a, b and c, 20×, 40× and 63× oil, respectively. (**C**) CD4 (brown)/Ki-67 (magenta) dual stain; panel a, low magnification (1×); panels b and c, 40× and 63× oil. (**D**) CD8 (brown)/Ki-67 (magenta) dual stain; panel a, low magnification (1×); panels b and c, 40× and 63× oil. (**E**) TIA-1; panels a and b, 20× and 63× oil; most of the atypical cells were TIA-1+. (**F**): Perforin; panels a and b, 20× and 63× oil; scant perforin-positive cells were identified. (**G**): FasL; panels a, b and c, 20×, 40× and 63× oil. (H): CD8; panels a and b, 20× and 63× oil. CCR4; panels c and d, 20× and 63× oil.

### 3.2. Morphologic and Immunohistochemical Features

All five biopsies exhibited similar morphologic and immunohistochemical features, usually with only minor intra- and inter-biopsy variation. The histologic features of the 2021 biopsy are detailed in Figure 2 and Figure 3. All biopsies displayed a highly vascularized atypical lymphoid infiltration that involved the dermis in the form of superficial perivascular and variably thick bands, with the formation of a *Grenz*-zone (Figure 2). Occasional deep dermal and subcutaneous fatty tissue involvement by perivascular infiltrates and a rare intraepidermal focus were noted (Figure 3A). The lesional infiltrate comprised small to medium-sized cells, with round, oval and moderately and markedly irregular (squiggly) nuclear contours, dense to irregularly clumped chromatin and indistinct nucleoli. Immunohistochemical studies showed the infiltrate to be positive for CD3, CD2, CD5, CD7 (mostly), TCR beta F1 and PD-1 (in greater than 30–50% of cells). Marked CD4 and CD8 immunostaining was noted. The pattern of TIA-1 expression was similar to that of CD8. Perforin (Figure 2F) and granzyme B highlighted scant cells. To further characterize the atypical cells, we stained them for FasL and CCR4 expression. We observed strong positivity of FasL and CCR4 in many of the atypical cells. Surprisingly, FasL was also strongly expressed in vascular endothelial cells (Figure 2G). Trapping of FasL and CCR4 positive atypical cells was observed in the walls of blood vessels (Figure 2G,H). CD68-KP1, CD68-PGM1 and lysozyme staining were limited to histiocytes (Figure 3). CD20 highlighted a few small B-cell clusters. CD30 highlighted occasional cells, estimated at 1.6–2.1% of cells. Ki-67 underscored a mostly mild proliferative fraction, estimated at up to 10%.

**Figure 3 biomedicines-11-00634-f003:**
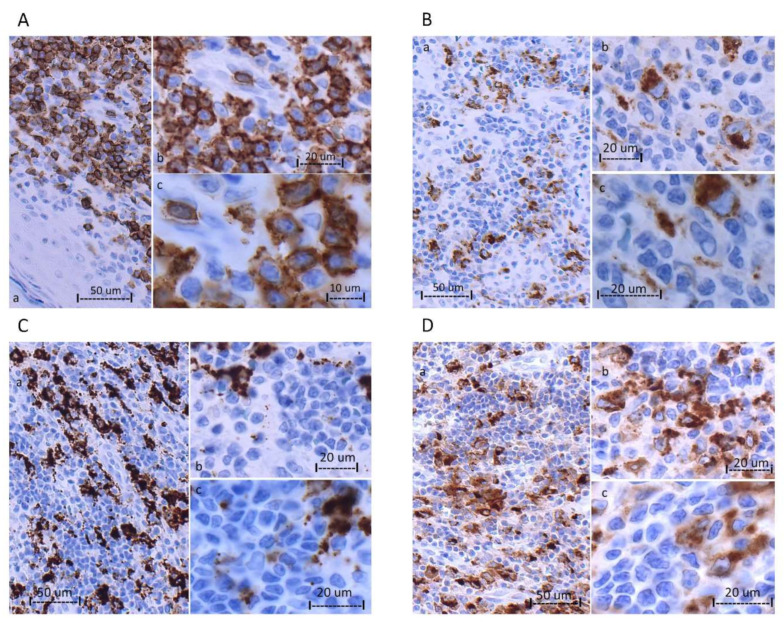
Photomicrographs of 2021 biopsy, histiocytic markers (**A**) CD8 highlights a rare epidermotropic focus. (**B**) CD68-KP1. (**C**) CD68-PGM1. (**D**) Lysozyme; panels a, b and c, 20×, 40× and 63× oil. Histiocytic markers only highlight histiocytes.

### 3.3. Flow Cytometric and Cell Sorting Analysis

Because of the confusing histological picture showing marked coverage for both CD4 and CD8 expression, we opted for flow cytometric (FCM) analysis of lymphocytes pooled from several of the patient’s skin lesions (Table 1, Appendix A). This confirmed the immunohistochemical impression that T-cells (CD3+) showed no loss of CD2 and CD5 and only a minor loss of CD7 (estimated at ~25%, in both CD4+ and CD8+ T-cell populations); no TCR gamma-delta expression was identified. CD279/PD-1 was measured at ~3.1% and 6.3% of CD4+ and CD8+ T-cell populations, respectively. The CD4:CD8 ratio was calculated at ~1:3.5 and 1:3.0 in the FCM analyses of the 2019 and 2021 biopsies, respectively (Table 1, Appendix A, panels a and b, respectively).

We proceeded to physically separate the CD4+ and CD8+ populations. The CD4 and CD8 T-cell populations were sorted using fluorescence-activated cell sorting (FACS) with the *FACSAria* Cell Sorter (BD Biosciences), according to the manufacturer’s instructions. As illustrated in Figure 4, our gating strategy isolated four pure populations, labeled as follows: CD4+/PD1-, CD4+/PD1+, CD8+/PD1- and CD8+/PD1+.

**Figure 4 biomedicines-11-00634-f004:**
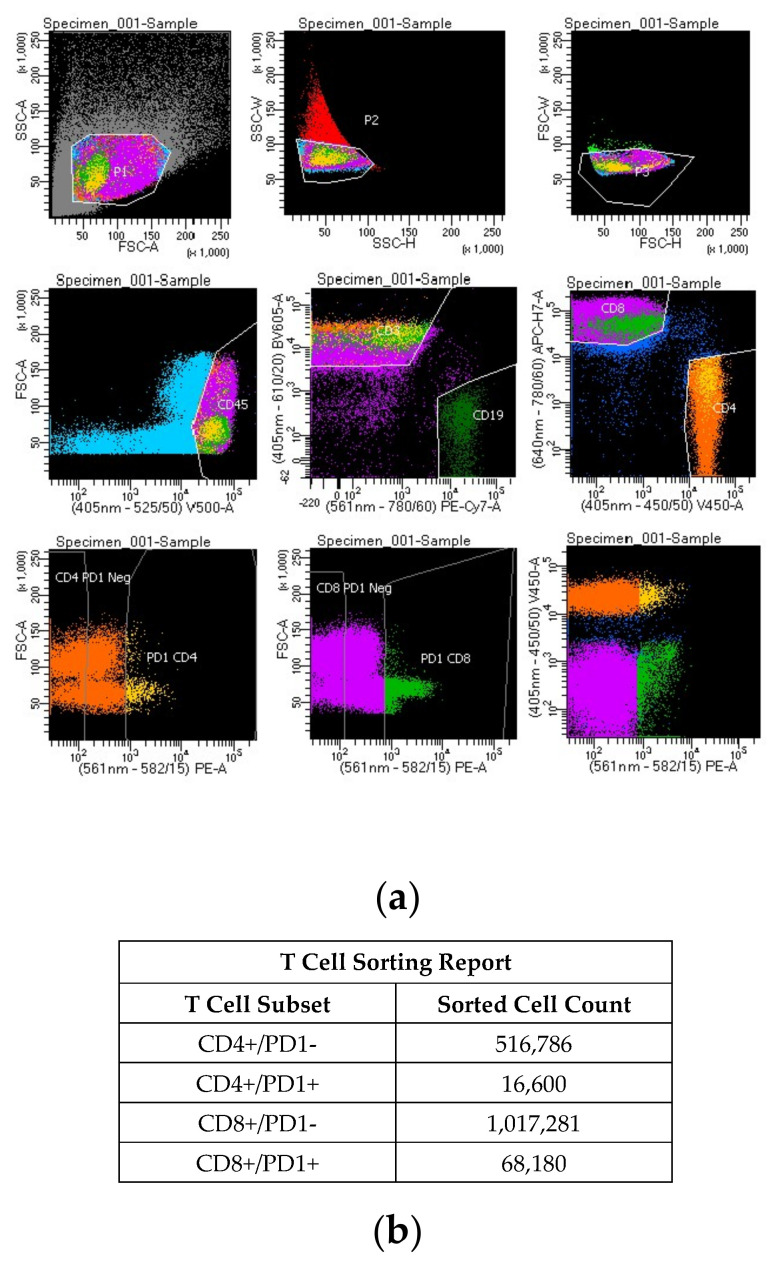
Legend: (**a**) Fluorescence-activated cell sorting (FACS); (**b**) cell sorter (CS) report.

### 3.4. T-Cell Receptor Clonality Analysis by Next-Generation Sequencing (NGS)

T-cell receptor (TCR) gene rearrangement studies for TCR gamma (TRG) and TCR beta (TRB) chains by NGS demonstrated the same clonal TCR gene rearrangements in CD8+ (PD1- or PD1+) sorted cell populations (Figure 5), which was the same as what was detected in the unfractionated cell extracts from the present as well as all prior skin biopsies that were performed at our institution. In contrast, a polyclonal pattern of TCR gene rearrangement was obtained in CD4+ (PD1- or PD1+) populations.

Figure 5 legend: T-cell receptor (TCR) gene rearrangement studies for TCR gamma (TRG) and TCR beta (TRB) chains (by NGS). Panel sets of two, from top to bottom, represent the following sorted cell populations: CD4+/PD1+ (AB), CD4+/PD1- (C,D), CD8+/PD1+ (E,F) and CD8+/PD1- (G,H).

### 3.5. Mutational Analysis for Potentially Drug-Targetable Mutations, Particularly JAK2

Because of the promising results of JAK2 inhibitor treatment of patients with JAK2 mutations, we investigated that possibility in our patient’s lymphoma. However, no JAK2 mutation was detected, either by an in-house myeloid NGS panel (that mainly detects JAK2 exon mutations) nor by the Foundation Medicine One (FM1) hematology panel, comprising more than 400 potentially drug-targetable genes (see end of Appendix A, Excel spreadsheet), including JAK2 fusion genes. Appendix A lists the detected variants (of undetermined significance).

## 4. Discussion

The case described herein was clinically characterized by a long, indolent course and multifocal cutaneous localization that presented on the legs, later involving the ears bilaterally, with multiple lesions on each ear. Immunohistochemical analysis showed the atypical infiltrate to be of T-cell lineage, positive for CD3, with marked CD4 and CD8 positivity. Molecular analysis revealed consistent clonal TRG and TRB TCR chain gene rearrangement. Cell sorting of the mixed population (pooled from multiple sites) followed by TCR analysis of purified CD4+/PD1+, CD4+/PD1-, CD8+/PD1+ and CD8+/PD1- populations showed, unequivocally, that the neoplastic population was CD8+, regardless of PD-1 expression status. Coupled with additional immunophenotypic findings, we demonstrate that the present indolent CTCL is caused by a neoplastic proliferation of cytotoxic T-cells that are CD3+ CD8+ CD4- TCRbF1+ TCRgd-. They exhibit a nonactivated cytotoxic phenotype, positive for TIA-1 but negative for other cytotoxic proteins (perforin and granzyme B). Moreover, they are negative for CD68 KP-1 and lysozymes. Many of the tumor cells strongly express FasL, which was strongly and diffusely expressed in the vascular endothelial cells of the lymphoma. Compared to recognized forms of morphologically similar types of CTCL, as described in the WHO’s classification of tumors of hematopoietic and lymphoid tissues, revised 4th edition, 2017 (updated in 2018) [1,3], we found that it differed from all of them, as discussed below.

**Primary cutaneous acral CD8+ T-cell lymphoma** (**PCACD8**) was first described as a possible phenotypic variant of the small-medium pleomorphic cutaneous T-cell lymphoma, arising on the ear, by Beltraminelli, Mullegger and Cerroni [11] and first coined by Petrella et al. and Greenblatt et al. [4,12], as an “indolent CD8+ lymphoid proliferation of the ear”. This was followed by a multicenter case series [5] which most closely resembled our case histologically and in immunohistochemistry, but there were significant differences. PCACD8 usually presents as a solitary lesion, most often in men (M:F ratio 3.2:1) and most frequently on the ear (61%). Other skin sites have been reported, namely the nose (22%), foot (8%) and eyelids, and anecdotally elsewhere. In our case, the first of multiple lesions on the ears did not appear until 2 years into our patient’s disease. The patient’s lesions began and mostly remained limited to her legs, and could be counted in the hundreds. In PCACD8, the immunophenotype is described as positive for CD3, CD8, β-F1 and TIA-1 and negative for CD4, CD20, CD30, CD56, granzyme B, PD-1, ICOS and CXCL13 [5,12], with a frequent weak expression or loss of other T-cell antigens, including CD2, CD5 and CD7 [4] and, rarely, CD4/CD8 co-expression [5]. However, no marked admixture of CD4+ cells that would obscure the identification of the neoplastic population has ever been described; moreover, a proportion of the neoplastic cells in our lymphoma expressed PD-1. Although lesions at other skin sites and the presence of multiple lesions have rarely been reported, they never been found in such great numbers; generally, only single or up to 2–4 lesions have been reported [6,11,13,14]. Because (Golgi dot-like) CD68 expression was shown to be a discriminative feature of PCACD8, distinguishing it from other CD8+ cutaneous lymphomas [6], we tested its expression in our case. However, no staining outside of admixed histiocytes was seen (Figure 3).

**Primary cutaneous gamma-delta T-cell lymphoma** (**PCGD**) may have a similar initial presentation of widespread cutaneous papules and nodules. However, immunophenotypically, our patient’s neoplastic cells were TCR alpha-beta positive (by immunohistochemistry) and not gamma-delta positive (by flow cytometry). Furthermore, PCGD did not match our patient clinically in that the former rapidly progresses to tumors and ulcerations, with a median survival of only 15 months [15,16]. Our patient had an indolent, purely cutaneous disease, present now for 7 years.

**Primary cutaneous CD8+ aggressive epidermotropic cytotoxic T-cell lymphoma** (pcAECyTCL) differs from our case in that it is characteristically epidermotropic, causes epidermal necrosis and shows a high proliferative fraction and aggressive clinical behavior, with a median survival of 12 months. Immunophenotypically, the two lymphoma types show some similarities, being CD3-, alpha-beta TCR-, CD8- and TIA-1-positive. However, pcAECyTCL shows an *activated* cytotoxic phenotype, with the expression of additional cytotoxic markers, including granzyme B and perforin, and the loss of some pan T-cell markers, including CD2 and CD5 [17,18,19,20].

In a seminal article by Bastidas et al. [21], deregulation of the JAK2 signal transduction pathway was shown to underlie pcAECyTCL. The lymphoma that we describe herein is neither aggressive nor epidermotropic. Those differences notwithstanding, we performed molecular genetics studies to investigate the possibility of a JAK2 gene mutation. However, no JAK2 exon or JAK2 fusion gene mutations were identified, either by an in-house myeloid neoplasms panel or by the comprehensive Foundation Medicine One (FM1) hematology panel of potentially drug-targetable gene mutations (end of Appendix A, Excel spreadsheet), lending further support to the possibility that overactive JAK2 signaling is a central driver of pcAECyTCL and leaving unanswered the question of which genetic mutations drive indolent CD8+ neoplasia.

**Primary cutaneous CD4+ small/medium-sized pleomorphic T-cell lymphoma/lymphoproliferative disorder** was the referred histological diagnosis, but this type of lymphoma presents as a solitary lesion in almost all cases [22,23]. In times past, rare cases with similar histopathology but with widespread skin lesions were included in this group. However, the current diagnostic WHO schema classifies such CTCL cases as peripheral T-cell lymphoma, NOS. Moreover, not only are such PTCL lesions widespread, but they involve rapidly growing tumors and exhibit greater than 30% large pleomorphic T-cells and/or a high proliferative fraction [24]. However, no such features were observed in our case. Most importantly, our case was CD8+.

An unexpected finding in our CTCL was the strong FasL expression seen in the tumor vascular endothelium. Selective strong FasL expression was reported by Motz et al. [25] in the vasculature of a number of solid human tumors, establishing their role as an immune barrier that promotes tolerance in such tumors. More recently, a possible role for FasL-expressing vascular endothelial cells as part of the mechanism of resistance to cancer immunotherapy by tumor infiltrating lymphocytes (TIL) was described [26]. However, to the best of our knowledge, there has been no scientific report of FasL overexpression in the vasculature of CTCL, such as that which we describe herein.

## 5. Conclusions

Primary cutaneous T-cell lymphomas (CTCLs) constitute a heterogeneous group of CLs that have highly characteristic clinical features, but often overlapping histologic features. As the armamentarium of personalized medicine increases, precise patient stratification becomes more valuable [27]. Herein, we described a case of a novel type of CD8+ CTCL that is characterized by indolent behavior, but marked multifocal presentation. We demonstrated how it differs from the entities currently listed in the gold standard CL classification schemata, employing clinical description, traditional histopathological and immunohistochemical methods and the cutting-edge technologies of cell sorting and molecular fingerprinting.

## Figures and Tables

**Figure 5 biomedicines-11-00634-f005:**
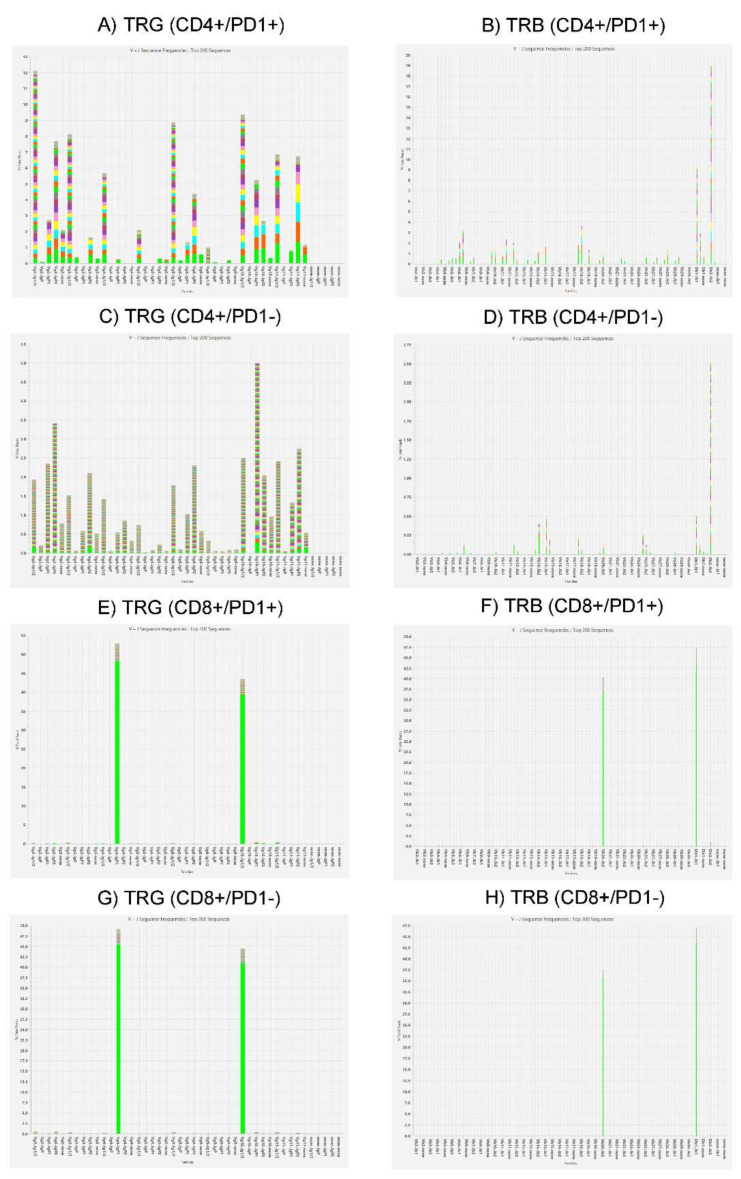
T-cell receptor clonality studies on 4 cell-sorted populations by next-generation sequencing (NGS).

**Table 1 biomedicines-11-00634-t001:** Biopsies procured and Treatments attempted.

Date	Comment
2016	Initial biopsy (left ankle and left thigh) procured at an outside institution, referred to as CSMP-TCL
November 2017	Radiation to several troublesome lesions (including a rare subungual one), with disappointing results; single session orthovoltage of 7 Gy
August 2017–December 2018	Isotretinoin up to 80 mg/m2 die (bexarotene unavailable, and alitretinoin limited access)—extreme xerosis without any benefit
January 2018	First biopsy at our institution (right anterior thigh—had been off treatment for 6 weeks at that point; clonal TCR+ for TRG and TRB; marked CD4+ and CD8+ by IHC
March–June 2018	Photochemotherapy: PUVA—minimal response after 12 weeks
August 2018	Repeat biopsy (right anterior thigh and right shin), same TCR+ for TRG and TRB; marked CD4+ and CD8+ by IHC IFN A2b 2.5 MU SQ TIW plus mechlorethamine, 0.01%, as a trial in left lower extremity only IFN stopped after 3 weeks because of adverse effects, prostrated by flu-like symptoms Nitrogen mustard stopped after 3 weeks due to adverse effects of intense local irritation
September 2018–April 2019	MTX 35 mg weekly (+FA)—no benefit
October 2019	Repeat biopsy (right thigh, 3 lesions, pooled)—had stopped MTX before this biopsy. Marked CD4+ and CD8+ expression by IHC. First of 2 FCM analyses of lymphocytes extracted from skin (Appendix A) showed 2 distinct T-cell populations, one CD4+ and one CD8+ (CD4:CD8, ~1:3.23); and the same TCR+ for TRG and TRB clonality pattern (by NGS)
April 2021	Repeat biopsy (right arm, right thigh, left thigh and left arm, pooled). Marked CD4+ and CD8+ by IHC (Appendix A). FCM (Appendix A) showed 2 T-cell populations, 1 CD4+ and 1 CD8+ (CD4:CD8, ~1:2.65). An unfractionated specimen showed the same TCR+ for TRG and TRB and marked CD4+ and CD8+ by IHC. Cell sorter analysis separated T-cells into 4 populations, 2 CD4+ (PD1- or PD1+) and 2 CD8+ (PD1a- or PD1a+). Each population was submitted for TCR clonality testing (by NGS), showing (identical) clonal TRG and TRB gene rearrangement in both CD8+, but in neither of the CD4+ T-cell populations (polyclonal)

Biopsies procured and treatments attempted are described. CSMP-TCL: CD4+ small/medium-sized pleomorphic T-cell lymphoma; TCR+: positive for clonal T-cell receptor gene rearrangement; TRB and TRB: TCR-beta and -gamma chain, respectively; IFN: interferon; MTX: methotrexate; MU SQ TIW: one million units subcutaneously three times/week; PUVA: psoralen + ultraviolet; IHC: immunohistochemistry; FCM: flow cytometry (BD *FACSCanto*, 10-color instrument); NGS: next generation sequencing. PD-1 (IHC) and CD279 (FCM): anti-PD-1 antibodies (terms used interchangeably).

## Data Availability

All data are included in the manuscript and Appendix A.

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
