# Peer review of "Primary Cutaneous Multifocal Indolent CD8+ T-Cell Lymphoma: A Novel Primary Cutaneous CD8+ T-Cell Lymphoma"

_biomedicines, 2023, doi:10.3390/biomedicines11020634_

Round 1

Reviewer 1 Report (Previous Reviewer 1)

The case report titled "Primary cutaneous multifocal indolent CD8+ T-cell lymphoma: a novel primary cutaneous CD8+ T-cell lymphoma" by Petrogiannis-Haliotis is an interesting analysis. Experiments are set up correctly, results are convincing, and the discussion is sufficiently in depth. The authors have also used adequate number of relevant references to support the ideas presented. 

Author Response

Thanks for your comment.

Reviewer 2 Report (New Reviewer)

Paper deals with important topics in Bioinformatics. The authors have reported the case of a patient, with a diagnosis of CD4+ small/medium-sized pleomorphic lymphoma.    

However, I have a number of suggestions:   

1. Abstract should be extended by the obtained results in part of the performance evaluation of the proposed approach entities described in 2018. 

2. Paper does not fit Bioinformatics journal guidelines.

3. Please optimize image sizing in the paper and table guidelines.  

4. The paper Result and Discussion chapter numbers are the same.  

5. The paper hasn’t had any Conclusion section.  

6. Please provide a link to an open-access repository with the dataset you attached at the end of the paper, and add them into references, it'll be a far better way, to organize it.

7. A lot of references are outdated and unlinked. Please fix it by using 3-5 years old papers in high-impact journals.

Author Response

The reviewer states that the manuscript must be improved notably in research design, adequate description of the methods used, presentation of the results, and support of the results by the conclusions. This is in complete contradiction to the comments of reviewer 1, first and revised manuscript, and of reviewer 2 of the first manuscript submission Our official answer: Research design, Methods used, presentation of Results, are cutting edge and up to date..   - Under « Comments and Suggestions for Authors » the new reviewer   2 refers to the case of our patient as being with diagnosis of CD4+  small/medium sized pleomorphic lymphoma. We emphasize that our lymphoma was a novel CD8+cutaneous lymphoma as named in the title and throughout the entire manuscript! Our official answer: To confound CD4+ and CD8+ CTCL in a manuscript review dealing with a novel CD8+ CTCL is a major error.
  - Under suggestions point 2. The reviewer mentions « Paper does not fit Bioinformatics journal guidelines ». We would like to remind everyone that « Bioinformatics » is an Oxford Academic journal that is distinct from Biomedicines. Our official answer: As written by the reviewer, it sounds like the journal Bioinformatics. We also point to the fact that Biomedicines is different from Bioinformatics.
>  - Under point 4. The reviewer says « Results and Discussion chapter  numbers are the same ». We carefully went over our manuscript again and could not confirm that statement (since there are no such numbers in the Discussion). Our official answer: This statement is incorrect.   - Under point 5. The reviewer mentions that the paper has no conclusion section. That simply is incorrect. There is a Conclusion section, at the end of the main body of the manuscript. Our official answer: This statement is incorrect. The Conclusion section was/is present.

 - Under point 7. The reviewer mentions that a lot of references are 
 outdated and unlinked. However, the other reviewers, including
reviewer 1 of the resubmission, mention an adequate number of
relevant references. We are confident that our reference choice is up
to date. If a group of authors has the unique chance to describe a
novel lymphoma, that group is obliged to compare its findings with
previously described (sometimes decades old) entities in the same
general category of neoplasms (in this case, primary cutaneous T-cell
lymphomas). Our official answer: When having the chance to describe a novel type of CD8+ CTCL, it is needed to compare with at a first glance similar, but after careful clinical and molecular analysis, different CTCL entities, implying "older" original descriptions. To cite only papers of the recent 3 to 5 years would be superficial.  

This manuscript is a resubmission of an earlier submission. The following is a list of the peer review reports and author responses from that submission.

Round 1

Reviewer 1 Report

The manuscript titled Primary cutaneous multifocal indolent CD8+ cutaneous T-cell lymphoma: a novel indolent primary cutaneous CD8+ T-cell lymphoma by Petrogiannis-Haliotis et al., is an interesting article. Adequate number of relevant references are used and the experimental design is sound. The only concern of this reviewer is that since the IHC revealed the CD8+ cells to be devoid of perforin/granzyme, the authors need to perfom additional studies to evaluate the expression profile of other cytotoxic molecules such as FasL, TRAIL, TNF alpha.  Since CD* cells are cytotoxic cells, evalaution of these cytotoxic molecules are very crucial.  

Reviewer 2 Report

In my opinion, in the presented form the manuscript (biomedicines-1869890) entitled ‘Primary cutaneous multifocal indolent CD8+ cutaneous T-cell lymphoma: a novel indolent primary cutaneous CD8+ T-cell lymphoma’ described by Tina Petrogiannis-Haliotis, Kevin Pehr, David Roberge, Ryan N. Rys, Yury Monczak, Gizelle Popradi, Lissa Ajjamada, Christiane Querfeld, Nathalie Johnson and Hans Knecht can be recommended for publication in Biomedicines in present form.

The text is comprehensible. The studies presented by the authors are interestingly described, properly and reliably documented by the novelty results. These results of the experiments regarding to novel type of CD8+ CTCL are presented, described and commented on at a high scientific level. Authors described a case of a novel type of CD8+ CTCL that is characterized by indolent behavior but marked multifocal presentation. In the study, authors have shown how it differs from entities currently listed in the golden standard CL classification schemata, employing clinical description, traditional histopathological and immunohistochemical methods and cutting-edge technologies of cell sorting and molecular fingerprinting.